# Do Recreationally-Trained Women of Different Ages Perceive Symptoms of the Menstrual Cycle and Adjust Their Training According to Phases?

**DOI:** 10.3390/ijerph192113841

**Published:** 2022-10-25

**Authors:** Isabella Righi, Renato Barroso

**Affiliations:** Department of Sport Sciences, School of Physical Education, University of Campinas (UNICAMP), Campinas 13083-851, Brazil

**Keywords:** perceptions, menstrual cycle phases, age group

## Abstract

We aimed to investigate the effects of the menstrual cycle (MC) in recreationally trained women athletes, including their perception of training, if age affected their perception of MC symptoms, and if they adjusted their training according to phases of the MC which they would perceive as the best/worst phase to train in. Three hundred- and ten-women amateur athletes with regular MC replied to an online quiz about their perception and the effects of MC on training and how they adjusted training according to their MC. Women were classified into three age groups: 18–25 years-old (n = 108), 26–35 years-old (n = 135), and 36–45 years-old (n = 67). Despite a higher ratio of younger perceived symptoms and the influence of MC phases in training, the group varied their training according to MC phases the least (37%) compared to 26–35 (50%) and 36–45-year-olds (40.2%). Most of athletes perceived the late follicular phase (LF) as the best phase to train in (18–25 = 79.6%; 26–35 = 80.7%; 36–45 = 91%) and the worst phases were early follicular (EF) (54.6%; 58% and 46.2%), and late luteal (LL) (38%; 48% and 47.7%). Regardless of age, most women perceived MC symptoms, and women in the 26–35 group adjusted their training more according to MC phases.

## 1. Introduction

The number of women participating in exercise and sport science research is lower than that of men [1]. It is suggested that the complexity of hormonal fluctuations, responsible for the menstrual cycle, and its possible influence on physiological responses to exercise [2] are to be blamed for the lower number of researchers investigating exercise in women and female athletes [1]. The menstrual cycle (MC) is a consequence of the physiological fluctuations of different hormones: the follicle-stimulating hormone (FSH) and luteinizing hormone (LH), estrogen, and progesterone. Although it is known that hormones affect how multiple systems of women’s organism function [3], results on the effects of hormones on exercise performance are still uncertain.

It is widely accepted that MC lasts, on average, 28 days, but the duration of a regular MC can vary between 21 and 35 days [4]. The MC is commonly divided into phases based on the fluctuation of the different hormones, especially estrogen and progesterone: in the early follicular stage (EF), both estrogen and progesterone levels are low; in the late follicular stage (LF), high estrogen levels are apparent; at ovulation (O) estrogen peaks; in the early luteal stage (EL) high levels of progesterone exist; at the mid-luteal level (ML)high levels of both estrogen and progesterone appear; and at the late luteal (LL) phase levels of estrogen and progesterone are low [4].

Even though the number of female participants in research is lower than that of men, the number of women participating in international sports competitions has increased over the past decades [5]. For instance, women participating in the Olympic Games represented less than 30% of the athletes 30 years ago in Barcelona in 1992 and increased to 48% in Tokyo 2020 [5]. However, training methods used by female athletes are similar to those used by men and do not take into account the possible influences of MC phases on women’s ability and willingness to train and if these influences affect adaptation to training. 

Although the physiological effects of MC on adaptations to exercise in female athletes are still debatable, as results of the investigations on this topic are still equivocal [2], some studies agree on how exercising women perceive MC symptoms [6,7,8,9,10]. In addition, MC symptoms have a potential to affect women’s ability to train. Accordingly, it has been suggested that women should adjust their training according to their MC phases [9]. Specifically, women should use higher intensities during the late follicular phase (LF) and in the early luteal phase (EL) and a lower intensity in the early follicular (menses) (EF) and late luteal phase (LL) due to the presence or absence of hormones (mainly estrogen and progesterone) [9,11]. However, these suggestions are not well established yet and do not account for possible age-group differences. 

The results of studies which have investigated the effects of aging in the MC symptoms of women indicate some aging-associated changes in MC. For instance, Barcelos et al. [12] reported that women aged between 35 and 44 years have a greater prevalence of MC disturbances (excessive menstrual bleeding and longer MC); on the other hand, Freeman et al. [13] showed that the severity of MC symptoms is worse in women in their late-twenties up to mid-thirties, and then decreases with aging. These findings suggest a possible association between age, perception of the MC symptoms and the influence of phases of MC on the ability and willingness to train. 

In 1931, it was first described that most women have complaints about symptoms of MC during the LL and EF phases [10]. These MC symptoms include, but are not limited to, irritability, feelings of depression, tension, and back pain. The perception of MC symptoms inversely relates to the ability to perform tasks of daily living but also extends to the ability to train and perform [6]. Although it has been known for nearly a century that most women feel MC symptoms, recently Armour et al. [14] reported the existence of an association between the perception of both the MC symptoms and the MC phases when women’s performance decreases. These authors investigated female athletes and observed that >80% of these athletes presented MC symptoms during the LL and/or EF phases, which coincided with their worst athletic performance [14]. Similarly, 17 elite athletes of different sports (athletics, climbing, weightlifting, gymnastics, judo) have the worst perceptions of MC symptoms in both LL and EF phases. In addition, these athletes reported higher levels of anxiety and distraction during competition, which may negatively impact performance. Motivation to train is also reduced due to physical disturbance and mood swings [9]. Thus, it is likely that MC phases and accompanying MC symptoms influence women’s training.

Most studies agree on the existence of MC symptoms during premenstrual (LL) and menstrual (EF) phases [6,7,8,9,10]; however, few studies have attempted to investigate if there is an association between the perception of MC symptoms and age. Additionally, as motivation to train is reduced in most exercising women, and there are negative effects of MC symptoms on their ability to perform, it is important to understand if and how women can adjust their training sessions during the phases when they feel better or worse. Therefore, the objective of this study was to investigate how exercising women of different age-groups perceive the symptoms of their MC and if and how they adjust their training sessions according to the MC phases.

## 2. Materials and Methods

### 2.1. Study Design

A 16-item online questionnaire containing questions about MC history, training experience, training frequency and duration, symptoms of MC, and adjustments in training due to MC phases, was posted and shared on social media. Participants were invited to provide answers to all of the questions.

The study procedures were approved by the Ethics Committee of the local University (approval protocol: 5.244.677), and participants were only allowed to see the questions after reading and accepting a consent form. 

### 2.2. Participants

Three hundred and thirty-five recreationally trained women participated in the study. Inclusion criteria included having a regular menstrual cycle (from 21 to 35 days) and being strength and/or endurance trained; exclusion criteria included using contraceptives, breastfeeding or being pregnant. 

Twenty-five responses were excluded (due to irregular MC and the use of oral contraceptives). The participants whose responses were considered valid were classified into three age groups: 18–25 (N = 108, 22.6 ± 2.3 years-old; 64.2 ± 16.4 kg; 1.64 ± 0.06 m; menarche age 12.0 ± 1.2 years), 26–35 (N = 135, 29.7 ± 2.9 years-old; 64.4 ± 11.1 kg; 1.63 ± 0.06 m; menarche 12.2 ± 1.5 years) and 36–45 (N = 67, 39.7 ± 2.4 years-old; 67.2 ± 11.4 kg; 1.64 ± 0.6 m; menarche 12.5 ± 1.4 years). 

### 2.3. Statistical Analysis

Data are presented according to descriptive statistics as means and standard deviation. The results of the questionnaires were presented in percentages. A Chi-square test was used to compare proportions. The significance was set at *p* < 0.05.

## 3. Results

We received 335 responses. However, after applying inclusion and exclusion criteria to the responses, 25 were discarded, and 310 were considered valid and used in the analysis. Regardless of the age group, the majority of participants presented symptoms of MC (99%). Specifically, 100% of the 18–25 age group, 98% of the 26–35 age group, and 99% of the 36–45 age group had pre-menstrual symptoms, with no difference between each age-group (χ^2^ = 2.35, *p* = 0.308). The most cited symptoms in each age-group are presented in Figure 1. Mood swings (84%), swelling (80%), irritability (73%), and cramps (72%) were the most cited symptoms and were felt by over two-thirds of the participants. Specifically, the main symptoms felt by women in each age-group were: 18–25 age group (92% mood swings; 81% swelling; 74% cramps; 74% irritability); 26–35 age group (81% mood swings; 77% swelling; 75% cramps; 72% irritability); 36–45 age group (83% swelling; 79% mood swings; 73% irritability; 64% cramps).

Figure 2 shows the percentage of women in each age group who perceived that the phases of MC affected their training: 99% of the 18–25; 97.7% of the 26-35; and 97% of the 36–45 age groups, with no difference between the age-groups (χ^2^ = 1.03, p = 0.598). Figure 2 below presents the relation between the symptoms, perceptions about MC phases and training, and training adjustments according to each age group.

We asked participants to answer which of the MC phases was the best and the worst to train in. The late follicular was indicated as the best phase to train in by 79.6% of women in the 18–25 age group, 80.7% in the 26–35 age group, and 91% in the 36–45 age group. The early follicular (menstruation) and late luteal (pre-menstrual) were pointed as the worst phases to train in by 54.6% and 38%, respectively in the 18–25 group, 58% and 48% in the 26–35 group, and 46.2% and 47.7% in the 36–45 group.

Thirty-seven percent of the 18–25 age group, 50% of the 26–35 age group, and 40.2% of the 36–45 age group adjusted their training according to the phases of MC, with no difference between the age groups (χ^2^ = 4.18, *p* = 0.124). The results regarding how these women adjust their training are presented in Table 1. Most exercising women decreased their training intensity (41.0%), while a substantial amount (17.2%) preferred not to train at all. The results for each age group are presented in Table 1.

## 4. Discussion

The purpose of this study was to investigate how recreationally trained women of different age-groups perceive the symptoms of their MC and if and how they adjust their training sessions according to the MC phases. The main finding was that almost all the participants in our study presented symptoms of MC, independently of age-group. Most women reported that MC had some influence on their training, but less than half of the women reported adjusting their training according to MC phases. 

Almost all the female rugby players (93%) [7], as well as Australian female athletes (83%) [14] investigated in these studies, reported MC symptoms. Although our numbers are slightly higher than previously observed (99% vs. 93% and 83%), the most cited symptoms by female athletes are similar: distraction [7], mood swings [7,14,15], increased appetite, water retention, lower back pain, and cravings [11]. The novelty of the present study is that, although non-significant, a slightly higher proportion of young adult women (18–25 years) reported feeling MC symptoms when compared with the 26–35 and 36–45 age groups. 

Although the main symptoms are similar in all the age groups (cramps, swelling, irritability, mood swings), the older group (36–45) seemed to perceive fewer cramps in comparison with other groups. Additionally, the 26–35 and 36–45 age groups indicated slightly fewer mood swings. These findings agree with those of Freeman et al. [13], who showed that the severity of MC symptoms decreases as women age. It is possible that women learn through experience what methods work to ameliorate the symptoms of MC. Interestingly, one of these methods, which was used by 42% in the study of Johnson, McChesney, and Bean [16], was exercise. Accordingly, aerobic exercise ameliorates symptoms of MC [17], and it is possible that long-term training has a cumulative effect on decreasing the symptoms of MC. Thus, as exercising women age, MC symptoms may become less severe.

Previous studies have observed that most women reported MC phases to affect their performance [7]. In our study, we were not interested in whether MC phases affected the participants’ performance; however, we were interested in whether participants felt that MC phases affected their training, which may include their ability and willingness to train. This is a novel finding. We observed that the proportion of women who felt that MC phases affected their ability or willingness to train was similar between women of different ages. Almost all the women in the younger age group reported that MC phases influenced their training, and slightly lower numbers were observed in the 26–35 and 35–45 groups. Although they may be associated, performance and ability or motivation to train are different, as a female may be unwilling to perform, but if she had to perform during a given MC phase, her performance would not be affected. Our findings contribute to the body of knowledge that prescribes training when taking into account women’s MC. However, care must be exercised by coaches, especially male coaches (who have never experienced MC symptoms), when having the information on MC phases and not using it against their female athletes.

In this study, women perceived the early follicular and late luteal as the worst phases to train in; late follicular, on the other hand, was considered the best. This is in agreement with one previous study [8] that showed that physical symptoms were presented immediately prior to or at the start of menses (the late luteal and early follicular phases), and psychological and affective symptoms were greater in the late luteal phase. This stands alongside the planning suggestion according to the phases of MC proposed by Pitchers and Elliot-Sale [9]. It is unknown if adjusting the training plan according to the phases of MC for women would affect performance. It is, however, conceivable that adjustments would make training more comfortable for menstruating women during some phases of their MC, which could have an impact on their willingness to train. For example, some type of exercise seems to ameliorate the symptoms, such as yoga and light aerobic exercise [18]. This is in agreement with our findings, especially from the 26–35 and 36–45 age groups, as participants reported skipping training sessions, decreasing their intensity and/or volume, and even changing the type of exercise. It is interesting that although women in the younger age group were those who reported the highest rate of perception in MC symptoms, they were also those who described fewer changes in training. Brown, Knight, and Forrest [8] bring up an interesting report from a female athlete, which seems to corroborate our findings. This athlete reports that she felt symptoms of MC but that there was no change in her training when she was younger because she did not understand her cycle [8]. Thus, it is likely that our younger participants feel less confident in or afraid of changing their training because it could have a negative impact on their future performance. Future studies are encouraged to investigate if modifying training according to women’s motivation and willingness to train during the early follicular and late luteal phases would affect their training adaptations and performance.

Our participants were recreationally trained practitioners, and we were unaware if they changed their training by themselves or if their coaches were responsible for it. Considering the fact that 75% of athletes [8] do not discuss menstruation with their coaches, we believe that participants changed their training sessions by themselves without their coaches’ knowledge. Oftentimes, athletes and coaches do not talk about how athletes feel because there is a barrier of communication [14], and some topics, such as menstruation, seem to be “forbidden”, especially with male coaches. Considering how communication may help to improve the empathy between coaches and athletes, and more empathic coaches could have a positive impact on athletes’ performance and success, we would like to encourage athletes and coaches to talk about MC, especially with the younger ones [19].

Highlighting this, our results have an important practical application to coaches and to amateur athletes. If the communication between coaches and athletes is improved, athletes may feel more open to talk to their coaches about MC phases and symptoms and how they are feeling. Coaches, being aware of how athletes are feeling, may adjust training sessions, making training more comfortable and avoiding incidences where athletes skip training. 

This study is not without limitations. First, this is a cross-sectional study where we asked participants about their perceptions of MC phases and symptoms, how they affect participants’ training and if participants changed their training in accordance with MC phases. Although we observed that some of the women reported training changes, we are uncertain if this practice would have long-term implications for the athletes’ performance. Second, we investigated recreationally trained athletes, and it is not clear whether the results are transferrable to an elite level. Third, we are unaware if all the respondents know about each phase of their MC. 

## 5. Conclusions

Regardless of age group, almost all recreationally trained women presented MC symptoms and perceived them as affecting their training. In spite of almost all our participants experiencing symptoms of MC, less than half of the participants in the 18–25 and 36–45 age groups and only half of the participants in the 26–35 age group adjusted their training according to the stages of MC. It seems that exercising women feel more comfortable either not training or decreasing their training volume and intensity. Although it is uncertain if modifying training according to MC phases would affect long-term adaptations and performance, nearly half of the participants felt the need to change their training.

## Figures and Tables

**Figure 1 ijerph-19-13841-f001:**
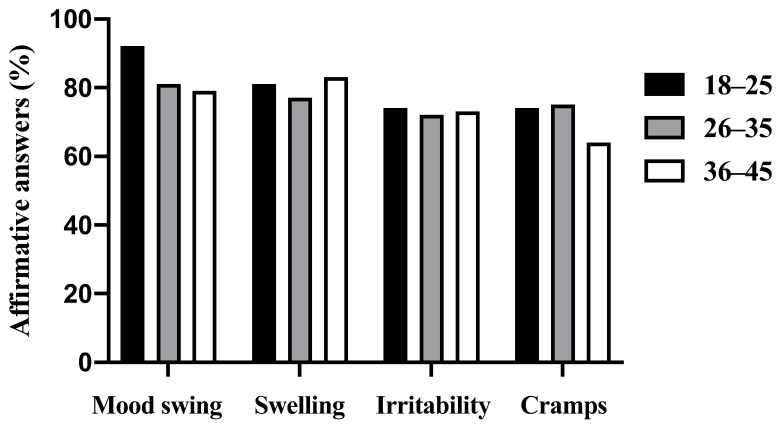
Four most-cited menstrual cycle symptoms according to age-group.

**Figure 2 ijerph-19-13841-f002:**
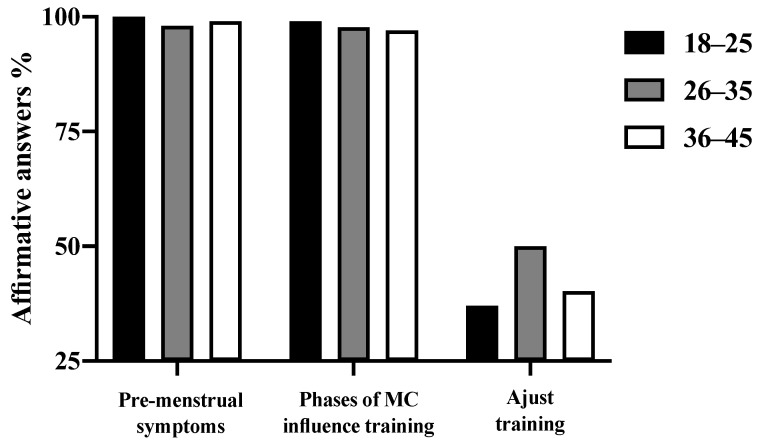
Percentages of women of different age-groups who reported feeling MC symptoms, who reported that MC phases influenced their training, and who indicated adjusting their training according to the MC phases.

**Table 1 ijerph-19-13841-t001:** Training adjustments according to their period with a symptomatic phase of MC (n = 134).

Change in Training	Age-Group		
	18–25 (n = 40)	26–35 (n = 67)	36–45 (n = 27)
Decrease intensity (%)	35.9	46.3	37.0
Do not train (%)	15.4	17.9	18.5
Decrease of intensity and volume (%)	15.4	11.9	29.6
Change the type of train (Different modality) (%)	10.3	7.5	7.0
Decrease volume (%)	2.6	7.5	7.4
Do not perform lower limb exercises (%)	2.6	1.5	0.0
Decrease frequency (%)	2.6	0.0	3.7
Did not answer (%)	10.3	6.0	0.0

## Data Availability

Data will be made available upon request to the authors.

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
