# Peer review of "Do Recreationally-Trained Women of Different Ages Perceive Symptoms of the Menstrual Cycle and Adjust Their Training According to Phases?"

_ijerph, 2022, doi:10.3390/ijerph192113841_

Round 1

Reviewer 1 Report

Dear Authors,

Thanks for your efforts in your research, however, your article is not eligible for publication in ijerph due to its subject, scope, content and originality. I have great and major hesitations about both the writing and the presentation of your research, so I have to reject your research with my most sincere regards.

Author Response

Thanks for your efforts in your research, however, your article is not eligible for publication in ijerph due to its subject, scope, content and originality. I have great and major hesitations about both the writing and the presentation of your research, so I have to reject your research with my most sincere regards.

Answer: We have rewritten some sections of the manuscript to improve readability and to accommodate reviewer's #2 suggestions. I hope it meets your expectations.

Reviewer 2 Report

This study investigated how recreationally-trained women perceive the symptom of the menstrual cycle and cope with this problem in regular sports training. The results in which the ages affect the perception of MC-related symptoms and the way of coping with this are interesting. Also, these findings are useful for promoting sports participation in women. However, I would better describe the rational explanation to raise the value of this study.  

#1. The latest research can not lead to the conclusion about MC-related performance alteration. In the Introduction, the first leading paragraph should be better described carefully.

#2. In the Introduction, I would better explain the rational reason why the authors focused on the AGE.

#3. In the Results, Figure 1 is a very interesting and important result. I would better show this information in each age. Also, I would better describe how and why the figure change across the age.

#4. In the Discussion, the authors should discuss the reasons for the age-specific difference in the perception of the symptoms of MC. The experience and knowledge about  MC would involve the perception and the way to cope with this.

#5. In the Discussion, I would better describe the clinical implication of the present results.

#6. Some references seem incorrect. In reference, I cannot confirm the Journal Name in the reference [2] and [14]. Please confirm the references.

Author Response

This study investigated how recreationally-trained women perceive the symptom of the menstrual cycle and cope with this problem in regular sports training. The results in which the ages affect the perception of MC-related symptoms and the way of coping with this are interesting. Also, these findings are useful for promoting sports participation in women. However, I would better describe the rational explanation to raise the value of this study.  

 Answer: We would like to thank the reviewer for their time and effort in reviewing our manuscript.  We read carefully the reviewer's comments and tried to accommodate the suggestion. We believe that their comments helped us improve our manuscript.

#1. The latest research can not lead to the conclusion about MC-related performance alteration. In the Introduction, the first leading paragraph should be better described carefully.

Answer: Thanks for the suggestion. We have rewritten the last sentence of this paragraph, and hope that it now meets the reviewer's expectation. 

#2. In the Introduction, I would better explain the rational reason why the authors focused on the AGE.

Answer: Thanks for the suggestion. We have added some information regarding age and menstrual cycle. 

#3. In the Results, Figure 1 is a very interesting and important result. I would better show this information in each age. Also, I would better describe how and why the figure change across the age.

 Answer: Thanks for the suggestion. We have split the result by age, but in order to improve readability we changed the type of figure to present this result. 

#4. In the Discussion, the authors should discuss the reasons for the age-specific difference in the perception of the symptoms of MC. The experience and knowledge about  MC would involve the perception and the way to cope with this.

 Answer: We have added some discussion to accommodate this suggestion.

#5. In the Discussion, I would better describe the clinical implication of the present results.

Answer: Thanks. We have added a paragraph describing the clinical implication of the results.

6. Some references seem incorrect. In reference, I cannot confirm the Journal Name in the reference [2] and [14]. Please confirm the references.

Answer: Thanks for pointing that out. We redid the references.

Round 2

Reviewer 1 Report

Congratulations. Your article is ready to publish.

Best.

Reviewer 2 Report

Thank you for your revised manuscript.

I have been satisfied with the author's revision.